# Early use of Antibiotics for at Risk CHildren with InfluEnza (ARCHIE): protocol for a double-blind, randomised, placebo-controlled trial

Kay Wang,[1] Tricia Carver,[1] Sharon Tonner,[1] Malcolm G Semple,[2] Alastair D Hay,[3] Michael Moore,[4] Paul Little,[4] Christopher Butler,[1] Andrew Farmer,[1] Rafael Perera,[1] Ly-Mee Yu,[1] Susan Mallett,[5] Jane Wolstenholme,[6] Anthony Harnden[1]

For numbered affiliations see end of article.

**Correspondence to**
Dr Kay Wang;
kay.wang@phc.ox.ac.uk

## ABSTRACT

**Introduction** Influenza and influenza-like illness (ILI) create considerable burden on healthcare resources each winter. Children with pre-existing conditions such as asthma, diabetes mellitus and cerebral palsy are among those at greatest risk of clinical deterioration from influenza/ILI. The Antibiotics for at Risk CHildren with InfluEnza (ARCHIE) trial aims to determine whether early oral treatment with the antibiotic co-amoxiclav reduces the likelihood of reconsultation due to clinical deterioration in these 'at risk' children.

**Methods and analysis** The ARCHIE trial is a double-blind, parallel, randomised, placebo-controlled trial. 'At risk' children aged 6 months to 12 years inclusive who present within the first 5 days of an ILI episode will be randomised to receive a 5-day course of oral co-amoxiclav 400/57 twice daily or placebo. Randomisation will use a non-deterministic minimisation algorithm to balance age and seasonal influenza vaccination status. To detect respiratory virus infections, a nasal swab will be obtained from each participant before commencing study medication. To identify carriage of potential bacterial respiratory pathogens, we will also obtain a throat swab where possible. The primary outcome is reconsultation in any healthcare setting due to clinical deterioration within 28 days of randomisation. We will analyse this outcome using log-binomial regression model adjusted for region, age and seasonal influenza vaccination status. Secondary outcomes include duration of fever, duration of symptoms and adverse events. Continuous outcomes will be compared using regression analysis (or equivalent non-parametric method for non-normal data) adjusting for minimisation variables. Binary outcomes will be compared using $\chi^2$/ Fisher's exact test and log-binomial regression.

**Ethics** The ARCHIE trial has been reviewed and approved by the North West-Liverpool East Research Ethics Committee, Health Research Authority and Medicines and Healthcare Products Regulatory Agency. Our findings will be published in peer-reviewed journals and disseminated via our study website (www.archiestudy.com) and links with relevant charities.

**Trial registration numbers** ISRCTN 70714783; Pre-results. EudraCT 2013-002822-21; Pre-results.

## Strengths and limitations of this study

► This randomised controlled trial specifically focuses on 'at risk' children with known risk factors for complications from influenza/influenza-like illness (ILI), who may potentially benefit most from early antibiotic treatment.
► The primary outcome (reconsultation due to clinical deterioration within 28 days) is clinically relevant and has important implications for cost-effectiveness of early antibiotic treatment.
► The study uses a pragmatic case definition of ILI, which reflects current clinical practice but does not reliably distinguish between influenza infection and ILI due to other causes.

## INTRODUCTION
### Background and rationale

Influenza is a viral infection that circulates mainly during winter and is a well-recognised risk factor for bacterial complications. 'At risk' children are more prone to becoming seriously unwell from influenza-related complications than otherwise healthy children.

'At risk' children are defined as children with underlying medical conditions or risk factors associated with an increased likelihood of developing complications from influenza/influenza-like illness (ILI). Based on guidance from the UK Department of Health[1] and the US Advisory Committee on Immunization Practices,[2] 'at risk' groups include patients with chronic respiratory, cardiac, renal, liver and neurological conditions, as well as diabetes mellitus and immunosuppression. There is also systematic review evidence, mainly based on data from children under 2 years of age, that premature birth (ie, delivery before 37 weeks' gestation) is a risk factor for influenza-related complications.[3]

BMJ

Based on data from Hospital Episode Statistics and the Office for National Statistics, 490 000 general practitioner (GP) consultations and 4200 hospitalisations due to seasonal influenza occur each year in children aged 14 years or younger.[4] This results in a cost to the National Health Service (NHS) of approximately £6.7 million due to hospitalisations[5] and £18 million due to primary care consultations.[6] The overall NHS and wider socioeconomic burden is likely to be greater due to additional costs incurred by critical care admissions, accident and emergency department attendances, clinical interventions (investigations and medications) and parental productivity losses (days off work and childcare costs).

Based on recommendations from the Joint Committee on Vaccination and Immunisation,[7] the UK introduced universal seasonal influenza vaccination for all children aged 2 and 3 years from 2013/2014, extending to all children aged 4 years during the 2014/2015 influenza season. In 2017/2018, Public Health England also initiated vaccination of children aged 4–5 years at school rather than in primary care and included children in year 4 of school (aged 8–9 years) in the childhood influenza vaccination programme.[8]

Vaccine uptake of 55.7% has been observed in areas using school delivery models, compared with only 34.7% when general practice delivery models are used.[9] However, vaccination rates in children with 'at risk' conditions have not improved from around 40% since 2013/2014.[10] Furthermore, effectiveness of the UK 2015/2016 seasonal influenza vaccination was only around 58% in children aged 2–17 years.[11] Evidence for treatment with neuraminidase inhibitors in 'at risk' children is also weak. Oseltamivir has not been demonstrated to provide significant benefit in terms of shortening symptom duration[12] or reducing acute exacerbations[13] in children with asthma.

Although there is a substantial evidence base underpinning recommendations that routine antibiotic treatment is not indicated for viral respiratory tract infections,[14] there is also preliminary evidence to suggest that early antibiotic use may be beneficial in preventing clinical deterioration and complications due to influenza. The results of a small, randomised, placebo-controlled trial suggest that early treatment with the antibiotic sultamicillin in children presenting with influenza/ILI significantly reduces the incidence of pneumonia.[15] Additionally, observational data from school-age children presenting in primary care with cough and fever found that duration of fever was significantly shorter in children with laboratory-confirmed influenza who received antibiotics (mostly amoxicillin) at an early stage during their illness.[16]

In recognition of the potentially serious clinical and socioeconomic consequences of bacterial complications of influenza, the UK government stockpiles the antibiotic co-amoxiclav for use during influenza epidemics and pandemics. Co-amoxiclav is the antibiotic of choice due to its coverage of both *Streptococcus pneumoniae* and *Staphylococcus aureus*, the most commonly observed bacterial coinfections in patients with laboratory-confirmed influenza.[17] In particular, *S. aureus* coinfections are associated with worse clinical outcomes in patients with laboratory-confirmed influenza[18] and were found in nearly half of influenza-related deaths between October 2004 and September 2012 among US children from whom at least one bacterial pathogen was identified from a normally sterile site.[19]

At the same time, antibiotic prescribing is a major driver of antimicrobial resistance,[20] which is well recognised as an emerging threat to the treatment of serious bacterial infections. Data on the potential impact of early co-amoxiclav use on development and duration of antimicrobial resistance are therefore needed to guide future national guidance on supply and use of this antibiotic during periods of high influenza activity. A clear, evidence-based understanding of the benefits versus harms will in turn support local antibiotic stewardship programmes and better informed, more appropriate decisions about antibiotic prescribing and use in the community.[21]

## METHODS AND ANALYSIS
### Objectives
Our primary objective is to determine whether early treatment with co-amoxiclav reduces the likelihood of reconsultation due to clinical deterioration in 'at risk' children with influenza/ILI.

Other objectives are:
► To determine whether early treatment with co-amoxiclav reduces duration of fever in 'at risk' children with influenza/ILI.
► To determine whether early treatment with co-amoxiclav reduces duration of symptoms in 'at risk' children with influenza/ILI.
► To compare further intervention rates in 'at risk' children with influenza/ILI treated with co-amoxiclav versus placebo.
► To compare adverse events in 'at risk' children with influenza/ILI treated with co-amoxiclav versus placebo.
► To assess the cost, outcomes and cost-effectiveness of early treatment with co-amoxiclav in 'at risk' children with influenza/ILI versus placebo.
► To determine the impact on long-term respiratory bacterial carriage and antibiotic resistance of early treatment with co-amoxiclav versus placebo in 'at risk' children with influenza/ILI.

### Study design
The ARCHIE trial is a multicentre, double-blind, randomised, placebo-controlled trial. 'At risk' children (ie, children with known risk factors for influenza-related complications), who present within the first 5 days of an ILI and who are not considered by their clinician to require immediate antibiotic treatment or hospitalisation, will be randomised to receive a 5-day course of co-amoxiclav 400/57 mg or a matching placebo. Nasal

**Box 1  'At risk' groups**

**Respiratory**
► Asthma requiring continuous or repeated use of controller therapy (eg, inhaled steroids, leukotriene receptor antagonists, long-acting beta agonists and systemic steroids).
► Admitted to hospital with exacerbation of asthma within the last 12 months.
► Admitted to hospital with bronchiolitis or pneumonia within the last 12 months.
► Recurrent viral wheeze (three or more episodes within the last 12 months).
► Bronchopulmonary dysplasia.

**Cardiac**
► Congenital heart disease being actively managed or monitored by cardiology team.
► Chronic heart failure being actively managed or monitored by cardiology team.

**Neurological**
► Chronic neurological or neuromuscular disorder that compromises respiratory function (eg, cerebral palsy).

**Renal**
► Chronic kidney disease defined as either of the following:
► Impaired estimated glomerular filtration rate* (eGFR) measurement within the last 12 months.
► Known hereditary or structural kidney abnormality with or without impairment in eGFR.
► Nephrotic syndrome.
► Kidney transplantation.

**Liver†**
► Cirrhosis.
► Biliary atresia.
► Chronic hepatitis.

**Immunodeficiency**
► Asplenia or splenic dysfunction.
► HIV infection.
► Undergoing chemotherapy leading to immunosuppression.
► Taking systemic steroids at a dose equivalent to prednisolone 20 mg or more per day (any age) or ≥1 mg per kg per day (children under 20 kg).

**Other**
► Diabetes mellitus (type 1 or type 2) or other metabolic condition.
► Genetic abnormality (eg, Down's syndrome).
► Sickle cell disease.
► Malignancy.
► Prematurity (born before 37 weeks' gestation) in children aged 6–23 months.

*Impaired eGFR is defined as an eGFR measurement of 59 mL/min/1.73 m$^2$ or less within the last 12 months before study entry. However, to enter the trial, the following two conditions must also be satisfied:
1. eGFR ≥30 mL/min/1.73 m$^2$ based on most recent measurement within the last 12 months.
2. No reason to suspect further deterioration in eGFR at time of study entry.

†Children with mild or moderate liver disease may enter the trial. However, to minimise the risk of serious hepatic complications related to study medication, children with severe liver disease may not enter the trial. Severe liver disease is defined as hepatic impairment associated with any of the following: jaundice, impaired coagulation/increased bleeding risk, bilirubin persistently greater than 50 µmol/L (two measurements within last 12 months).

swabs will be obtained from all participants to detect influenza infection. Throat swabs for culture and sensitivity will also be obtained at baseline where possible. The primary outcome is reconsultation due to clinical deterioration within 28 days of randomisation.

## Study participants

We will recruit 'at risk' children aged 6 months–12 years inclusive who present within the first 5 days of an ILI. We will define ILI as the presence of both cough and fever, which may be defined as child-reported fever, parent-reported fever or temperature >37.8°C (axillary or tympanic temperature measurement). This is intended as a pragmatic case definition, which can be reliably applied across our entire target age range.

Box 1 summarises our definition of 'at risk' groups. Rather than being an exhaustive list, this definition is intended to guide clinicians in identifying which children are likely to be at greater risk of influenza-related clinical deterioration or complications. Healthcare professionals are also advised to use their clinical judgement to identify 'at risk' children. Children with other potential risk factors who may be suitable to take part may be discussed with a medically qualified member of the research team. Children who require immediate antibiotics or hospital admission for treatment of an influenza-related

**Box 2    Study inclusion and exclusion criteria**

**Inclusion criteria**
► Aged 6 months–12 years inclusive.
► In 'at risk' category.
► Presenting with influenza-like illness (ie, cough and fever) during influenza season.
► Presenting within 5 days of symptom onset.
► Permanently registered at a general practice in UK.
► Parent/guardian able to complete study diary and questionnaires.

**Exclusion criteria**
The participant may not enter the trial if ANY of the following apply:
► Known contraindication to co-amoxiclav.
► Child given antibiotics for treatment of an acute infection within the last 72 hours.
► Child requires immediate antibiotics (clinician's judgement).
► Child requires immediate hospital admission for treatment of an influenza-related complication (clinician's judgement).
► Child has been observed on hospital ward or ambulatory care unit for longer than 24 hours.
► Presence of any reason to prevent healthcare professional from obtaining nasal swab.
► Child with known cystic fibrosis.
► Child previously entered into the Antibiotics for at Risk CHildren with InfluEnza trial.
► Child has been involved in another medicinal trial within the last 90 days.

complication (based on their clinician's judgement) will not be eligible to take part. Box 2 summarises our full inclusion and exclusion criteria.

### Recruitment

We will recruit study participants from a range of healthcare settings where 'at risk' children from the community present with influenza/ILI. Recruiting sites will include general practices, walk-in centres and hospitals. Identification of participants at these sites may be supported by the use of participant identification centres. Given the seasonal nature with which influenza circulates in the community, each recruitment season will commence at the beginning of October and continue until the end of March the following year or later if influenza continues to circulate above seasonal thresholds.

Before each recruitment season, we will ask participating general practices to perform database searches to identify children within our target age range who are in 'at risk' groups. A medically qualified individual will manually screen the results of the database search to exclude any children who would not be suitable to invite to take part in the study. Recruiting sites may also install a study reminder screen prompt, which will appear when the electronic medical records of potentially suitable children are accessed. Participating hospitals will also be asked to identify potential recruits through specialist outpatient clinics that manage children with 'at risk' conditions.

To further raise awareness about the trial and opportunities for participation, we will provide recruiting sites with study promotional materials including posters and short information leaflets, which summarise our study aims and direct families of potential participants to our study website (www.archiestudy.com). We will also liaise with recruiting sites to raise awareness via local news outlets, social media and charities representing relevant patient groups. Parents who express an interest in allowing their child to take part in the study will be able to contact the study team or look at the study website to find their nearest recruiting site.

During each recruitment season, children who present at recruiting sites with influenza/ILI will be screened to determine whether they are eligible to take part in the trial. Suitably trained healthcare professionals will then obtain informed consent for study participation from the child's parent or guardian and complete study enrolment and randomisation procedures either at the recruiting site or at the child's home within 24 hours of confirming their eligibility for study participation.

### Intervention

Participants will be randomised to receive either co-amoxiclav 400/57 (amoxicillin 400 mg as trihydrate and clavulanic acid 57 mg as potassium salt/5 mL when reconstituted with water) or a matching placebo, which will be taken orally twice daily for 5 days. Study medication doses based on age, as well as weight in participants aged 6–23 months, will be calculated according to the standard dosing regimen for co-amoxiclav 400/57 recommended by the British National Formulary (BNF; see table 1). Study medication will be provided as a dry powder, which healthcare professionals will reconstitute with water once the participant has been randomised.

Medically qualified individuals will use their clinical judgement when advising on study medication doses for any children to whom they feel that standard BNF dosing recommendations should not apply.

Participants will be allowed to continue their usual regular medications and any additional medications for their influenza/ILI episode while taking part in the trial.

**Table 1**    Study medication dosing regimen

| Child's age | Study medication dose |
| --- | --- |
| 6–23 months | |
| Under 6 kg | Calculate dose according to British National Formulary instructions for co-amoxiclav 400/57. Advise two doses daily for 5 days. |
| 6.0–7.9 kg | 1 mL twice daily for 5 days. |
| 8.0–10.9 kg | 1.5 mL twice daily for 5 days. |
| 11.0–12.9 kg | 2 mL twice daily for 5 days. |
| 2–6 years | 2.5 mL twice daily for 5 days. |
| 7–12 years | 5 mL twice daily for 5 days. |

## Baseline study procedures

Healthcare professionals will record baseline data from study participants on age, sex, comorbidities, household smoking status, influenza vaccination status, medications given during the current illness episode, duration of fever and duration of symptoms. Heart rate and respiratory rate will also be measured and recorded at baseline. Other baseline characteristics (medical conditions, regular medications, vaccinations and acute consultations that occurred up to 12 months before randomisation) will be extracted from participants' medical records.

A nasal swab will be obtained from each participant at baseline. Nasal swabs will be placed in viral transport medium and sent by post to Alder Hey Children's Hospital microbiology department for analysis, where they will be analysed by real-time PCR analysis to detect influenza and distinguish influenza A, B and A/H1N1 2009 pandemic subtypes. Residual medium will be retained for potential future detection of other pathogens. Where possible, a throat swab will also be obtained from the participant for culture and sensitivity. In a subgroup of participants whose parents/guardians give additional written informed consent, further throat swabs will be collected after 3, 6 and 12 months. Online supplementary appendix A provides further details of the laboratory analyses, which will be performed on throat swab samples.

## Outcomes

The primary outcome is reconsultation due to clinical deterioration within 28 days of randomisation. Clinical deterioration is defined as any of: worsening symptoms, development of new symptoms or development of a complication requiring medication or hospitalisation. This definition is based on that used by the Genomics to combat Resistance against Antibiotics in Community-acquired lower respiratory tract infection in Europe consortium in order to define reconsultations due to clinical deterioration in patients with lower respiratory tract infections.[22]

Secondary outcomes include duration of fever and duration of symptoms from time of randomisation and medication prescriptions and/or further investigations, adverse events, hospitalisations or deaths, all within 28 days of randomisation. For the purpose of the economic evaluation, outcomes include healthcare resource utilisation and parental/informal care costs within 28 days of randomisation and health-related quality of life as measured by the EuroQol health-related quality of life youth proxy version (EQ-5D-Y proxy) at baseline and days 4, 7, 14 and 28.

## Data collection

Online supplementary appendix B presents a complete summary of data collected during the trial. Data obtained at the baseline assessment, week 1 and week 2 follow-ups, as well as data on safety events, will be entered on paper case report forms before being sent to the main trial office for entry using OpenClinica open source software (V.3.13;

Waltham MA, USA), a validated online electronic data capture system. All data will be double entered to ensure accuracy. Data extracted from the medical notes of children recruited from general practices will be directly transcribed into OpenClinica at the child's general practice. The main trial office will contact the general practices of children recruited from hospitals or walk-in centres to request relevant data from their medical notes, which will then be entered by a member of the trial team.

A parent or guardian will be asked to complete and return by post a series of four 1-week study diaries on behalf of the participant to provide data on doses of study medication given to the child, axillary temperature, symptoms, adverse events, daily activities and childcare, child's health-related quality of life measured by the EQ-5D-Y proxy and health service contacts such as hospitalisations and visits to the GP. Healthcare professionals will also contact each participant's parent or guardian by telephone 1 week and 2 weeks after randomisation to collect data on duration of fever, adverse events and adherence to study medication. Primary outcome data will be extracted from participants' medical notes at least 3 months after randomisation. This is to allow records of any hospital admissions or consultations in non-general practice settings to be captured in the child's general practice medical record. Data on duration of symptoms will be collected from study diaries completed by participants' parents/guardians. Data on duration of fever will also be collected from study diaries and supplemented by follow-up telephone calls with parents/guardians 1 week and 2 weeks after randomisation. During these telephone calls, parents/guardians will be asked whether their child's fever has settled and, if so, when. Data on adverse events will also be collected at these telephone calls. Data on medication prescriptions, further investigations, hospitalisations and deaths will be extracted from participants' medical records.

## Randomisation and blinding

Randomisation will be stratified by region with minimisation for age (6–23 months inclusive versus 2–12 years inclusive) and seasonal influenza vaccination status (yes or no/do not know). Randomisation is performed using Sortition, an online clinical trial randomisation software package developed by the Clinical Trials Unit at the University of Oxford, Nuffield Department of Primary Care Health Sciences. The randomisation system will be implemented and managed by the Clinical Trials Unit. The randomisation codes will be maintained by a specifically appointed independent custodian. Healthcare professionals, the study team, children and parents/guardians will all be blinded to treatment allocation.

A participant's treatment allocation will be unblinded in the event of a suspected unexpected serious adverse reaction. Where there is a perceived need for unblinding, the participant's responsible clinician (ie, the clinician treating the participant) will discuss the case with the chief investigator or a designated alternative study clinician. If

it is agreed that unblinding is required, the chief investigator or designated study clinician will request that the randomisation code is accessed by the independent custodian, who will disclose the participant's treatment allocation to his or her responsible clinician.

## Sample size

A large population-based study using the UK General Practice Research Database found that true complications occurred in 17.6% of at-risk children aged 1–14 years within 30 days of being clinically diagnosed with influenza/ILI.[23] Assuming that true complications account for 44% of reconsultations due to clinical deterioration,[24] we estimate that 40% (17.6%/44×100) of at-risk children with clinical influenza will reconsult with clinical deterioration within 30 days of initial presentation.

We will aim to recruit 650 children into the trial, including an inflation factor of 1.041 and allowance for 25% loss to follow-up. Our target effective sample size is therefore 484 children (242 in each arm), which would be sufficient to detect a reduction in reconsultation due to clinical deterioration from 40% to 26% (a 35% relative risk reduction) with 90% power and 5% two-tailed alpha error. Our inflation factor estimate is based on a conservative intracluster correlation estimate of 0.03[25] and a coefficient of variation of 0.6 (based on the value observed in the Diabetes care from Diagnosis (DD) trial),[26] assuming an average cluster size of two patients.[27]

We acknowledge that recruitment of the above sample size is likely to be challenging (see Discussion). However, a sample of 280 participants would still be sufficient to detect a reduction in reconsultation due to clinical deterioration from 40% to 23% with 80% power and 5% two-tailed alpha error, including an inflation factor of 1.041 and allowance for 5% loss to follow-up. This effect size (a 42.5% relative risk reduction) is still conservative compared with the effect observed in the aforementioned trial conducted by Maeda et al,[15] which reported an 85% relative risk reduction in incidence of pneumonia in children with ILI who were treated with the antibiotic sultamicillin (1/42 children) versus placebo (7/43 children). A 5% loss to follow-up rate should also be achievable given that data on reconsultations due to clinical deterioration will be extracted from medical notes.

## Statistical analysis and economic evaluation plan

We will perform an intention-to-treat analysis including all randomised participants and use multiple imputation methods to account for missing data. Specifically, the participants will be analysed in the groups to which they were allocated. Baseline characteristics will be summarised by treatment groups. The results from the trial will be presented as comparative summary statistics (difference in proportion or means) with 95% CIs. The analysis and reporting of results will follow the general principles of Consolidated Standards of Reporting Trials 2010 statement.[28]

The proportion of children reconsulting due to clinical deterioration in the two groups (primary outcome) will be compared using a log-binomial regression model with adjustment for region, age and seasonal influenza vaccination status. The treatment effect will be reported as a relative risk with 95% CI. A p value will also be presented. Stability and assumptions of the regression model will be explored, and alternative method will be used if any violation of assumptions occurred.

Analysis of continuous outcomes (eg, duration of fever and duration of symptoms) will be compared using regression analysis, adjusting for the same factors as described above. A non-parametric method will be used if assumptions of linear regression are not met. Binary outcomes (eg, proportions of children prescribed medication and/or requiring further investigations, children in whom adverse events are reported and children who are hospitalised or died within 28 days of randomisation) will be compared using $\chi^2$/Fisher's exact test and log-binomial regression. Sensitivity analyses will be carried out to examine the robustness of the results with different assumptions about departures from randomisation policies and handling of missing data.

The incremental costs of administering early treatment with co-amoxiclav in 'at risk' children with influenza/ILI versus placebo will be estimated using the data on healthcare resource use provided by the four 1-week study diaries and the medical notes review. We will estimate total costs and costs relating to burden on primary care, secondary care and parental/informal care. We will extrapolate our analysis of resource use and costs to explore the potential cost impact of early co-amoxiclav use on a national scale. The primary outcome measure for the cost-effectiveness analysis will be the EQ-5D-Y proxy. We will estimate and report all the costs and consequences in a disaggregated format (cost–consequences analysis) as well as analysing and reporting the incremental cost and effectiveness in terms of cost per quality-adjusted life year of administering co-amoxiclav versus placebo alongside a sensitivity analysis.

## Safety and adverse event reporting

We will record all adverse events notified by trial investigators, healthcare professionals or participants as occurring within 28 days of randomisation, whether they are related to study medication. However, we will not undertake formal adverse event reporting for events which are known to be common side effects of co-amoxiclav (ie, mucocutaneous candidosis, diarrhoea, nausea and vomiting)[29] unless they result in a serious adverse event or are considered by the child's clinician to be clinically severe.

Clinicians will be advised to discontinue a participant's study medication if he or she experiences an adverse drug reaction related to the study medication. Parents/guardians of participants whose study medication is discontinued will still be requested to complete their study diaries and questionnaires and will still receive telephone

follow-up calls unless they choose to withdraw consent for these.

The independent Data and Safety Monitoring Committee (DSMC) for the trial will be responsible for reviewing safety events after each recruitment season. The main aims of this review are:

► To ensure the safety of each trial participant.
► To identify any trends, such as increases in unexpected adverse events, and take appropriate action.
► To seek additional advice or information from investigators where required.
► To evaluate the risk of the trial continuing and take appropriate action where necessary.
► To act or advise, through the chairperson or other consultant, on incidents occurring between meetings that require rapid assessment.

The DSMC will also advise on whether the trial should be terminated based on its reviews of serious adverse events in consultation with the Trial Steering Committee and trial sponsor if necessary.

## Patient and public involvement

Our patient advisers and patient representatives identified via the National Institute for Health Research (NIHR) Medicines for Children Research Network Clinical Study Groups helped develop and refine our study objectives by providing extensive feedback on early draft versions of our research plan. To inform the design of our trial materials and procedures, we held discussion groups with 21 parents of children with 'at risk' conditions and 15 young people in four different locations (Oxford, London, Liverpool and Birmingham). These consultations informed the design of our 'Archie the penguin' mascot, study information leaflets and study diaries (four 1-week diaries instead of one diary to be completed over 4 weeks), as well as selection of the EQ-5D-Y proxy questionnaire to measure health-related quality of life in trial participants. They also prompted us to raise awareness about the study before the start of each recruitment season, produce a series of short videos on our study website to explain the study to parents and guardians and initiate set-up of the home visit recruitment model in Clinical Research Networks (CRNs) with the resources to support this. Patient representatives will also help us disseminate our research findings by providing assistance with writing plain language summaries and establishing links with relevant charities and patient organisations that represent parents and children to whom the findings of our research will be relevant.

## DISCUSSION
### Implications of study findings

The findings of the ARCHIE trial will provide important evidence to inform clinically appropriate and cost-effective antibiotic prescribing and use in 'at risk' children presenting with ILI in community healthcare settings. The only randomised placebo-controlled trial conducted to date of early antibiotic use in children with ILI[15] excluded children with chronic medical conditions (even though these children are at greatest risk of clinical deterioration) and did not evaluate potential implications of antibiotic treatment on antimicrobial resistance. There are also no studies to date that have examined the cost-effectiveness, costs and effectiveness of early antibiotic use in 'at risk' children with influenza/ILI.

The lack of clear, evidence-based guidance in this area results in considerable uncertainty and wide variation in clinical decision making about prescribing antibiotics to children with ILI who have known risk factors for influenza-related clinical deterioration.[30] This may in turn result in antibiotics being prescribed unnecessarily, leading to emergence of antibiotic-resistant infections,[31] which are associated with worse clinical prognosis, even in patients with common respiratory tract infections in the community.[32]

## Reflection

The ARCHIE trial is now in its fourth and final recruitment season. However, recruitment has proven to be more challenging than originally anticipated due to a significantly delayed start in recruitment, a limited pool of potential recruits and limitations in capacity to recruit at participating sites.

We had originally planned to start recruitment in October 2014 and continue recruitment over three winter seasons (2014/2015, 2015/2016 and 2016/2017). However, we were only able to start recruitment on 11 February 2015 due to delays in obtaining our study medication. By this time, the peak period of influenza/ILI consultation activity in early January 2015 had already passed.[33]

Extension of the UK childhood universal seasonal influenza vaccination programme to all children aged 4 years during winter 2014/2015 and subsequent annual extension of that programme to older children may also have had an adverse impact on recruitment. Indeed, primary care consultation rates for influenza/ILI in England peaked at relatively modest levels during winters 2015/2016 (28.7 per 100 000)[34] and 2016/2017 (20.3 per 100 000)[35] compared with previous influenza seasons. The 2016/2017 influenza season was particularly mild, with influenza/ILI consultation rates in England remaining above the Moving Epidemic Method baseline threshold of 14.3 per 100 000 for only 6 weeks.[35] However, previous research suggests that GPs rarely consider influenza infection or ask about vaccination status when assessing 'at risk' children with ILI.[30] It is therefore unlikely that recruitment would have been significantly compromised due to healthcare professionals failing to consider vaccinated children as potential recruits.

The highly specific nature of our study eligibility criteria also means that recruitment is only possible from a limited pool of children with one or more known 'at risk' conditions (see box 1). Additionally, the healthcare professional assessing the child must be satisfied that the

child does not require immediate antibiotic treatment. This has so far been a major barrier to recruitment, with the most common reasons for exclusion at the screening stage being receipt of antibiotics within the last 72 hours or requirement for immediate antibiotics. This suggests that parents and/or healthcare professionals were identifying specific reasons to treat these children with antibiotics, or erring on the side of caution by initiating antibiotic treatment in cases when they were unsure about clinical prognosis in the absence of treatment.

Staff at both primary care and hospital ambulatory care recruiting sites have reported difficulties with finding time to recruit participants alongside other work commitments at a time of increased seasonal health seeking activity. Due to the nature of ILI, it is often not possible to anticipate when these children will present. Parents and children are sometimes unwilling to wait until suitably trained recruiting staff become available. Four-hour waiting time targets in hospital emergency departments mean that capacity to recruit is particularly limited in those settings, especially if there is no alternative space in which children can be assessed.

### Strategies for addressing challenges encountered

To increase our pool of potential recruits, and compensate for our delayed start in recruitment, we have increased our total number of recruiting sites as far as staff capacity and resources will allow. To support efficient training of recruiting staff, we have produced a series of training videos on eligibility assessment, baseline clinical assessment, nasal and throat swab techniques, randomisation, study questionnaires and diaries, follow-up assessments (including adverse event reporting) and medical notes reviews, which can be accessed via our study website. Study expansion has also been facilitated by utilisation of existing infrastructure within the NIHR CRNs to set up a home visit model for recruitment from general practices.

Recruitment via the home visit model started in winter 2016/2017. This model only requires general practice staff to identify potentially eligible children before referring them to the study team, who then arrange for a research nurse to obtain informed consent and recruit children in their homes within 24 hours of initial presentation. This is therefore much less burdensome to individual general practices than recruiting participants on site.

We have also used CRN infrastructure to identify additional hospitals as recruiting sites, particularly focusing on hospitals with children's assessment units and direct access pathways that allow families of children with known 'at risk' conditions to bring children directly to the children's assessment unit or ward instead of to the emergency department.

Additionally, we made three minor changes to our eligibility criteria in advance of starting recruitment in winter 2017/2018. Our inclusion criteria now state that children who are permanently registered at a general practice in the UK (rather than in England only) are potentially eligible to participate. This has allowed us to set up the

ARCHIE trial in Wales, with the support of the Health and Care Research Wales Support Centre, which has experience of home visit recruitment for primary care clinical trials.[36] We have also clarified that children who have received antibiotics during the last 72 hours will only be excluded if these antibiotics were given to treat an acute infection. Children on long-term antibiotic prophylaxis are therefore still potentially suitable for inclusion in the study. Additionally, we have clarified that children who are felt to require immediate hospitalisation will only be excluded if this is specifically for treatment of an influenza-related complication or for a period of observation lasting longer than 24 hours.

### Ethics and dissemination

The ARCHIE trial is registered on the International Standard Randomised Controlled Trials registry (70714783) and European Clinical Trials Database (EudraCT: 2013-002822-21). We have obtained full ethical approval for this study from the North West – Liverpool East Research Ethics Committee (13/NW/0621), as well as approvals from the Health Research Authority and Medicines and Healthcare Products Regulatory Agency. All relevant amendments have been assessed and approved by Research Ethics Committe/Health Research Authority and Medicines and Healthcare products Regulatory Authority Local research and development approvals/confirmation of capacity were received at all recruiting sites before recruitment commenced.

The trial will be conducted in accordance with the principles of Good Clinical Practice and relevant regulations. Full written informed consent will be obtained from a parent or guardian for each study participant. Children will also be invited to give written assent if appropriate. To ensure that their confidentiality is maintained, participants will be identified only by a participant identification number on all Case Report Forms apart from the contact information sheet, consent form and assent form (if completed). All documents will be stored securely and will only be accessible by trial staff and authorised personnel. The study will comply with the Data Protection Act, which requires data to be anonymised as soon as it is practical to do so.

Our findings will be published in peer-reviewed journals and disseminated to patients and the public via our study website (www.archiestudy.com) and links with relevant charities representing children with 'at risk' conditions.

**Author affiliations**
[1]Nuffield Department of Primary Care Health Sciences, University of Oxford, Oxford, UK
[2]Women's and Children's Health, Institute of Translational Medicine, University of Liverpool, Alder Hey Children's Hospital, Liverpool, UK
[3]Centre for Academic Primary Care, Population Health Sciences, Bristol Medical School, University of Bristol, Bristol, UK
[4]Academic Unit, Primary Care and Population Sciences, University of Southampton, Aldermoor Health Centre, Southampton, UK
[5]Institute of Applied Health Research, University of Birmingham, Birmingham, UK

[6]Health Economics Research Centre, Nuffield Department of Population Health, Oxford University, Oxford, UK

**Acknowledgements** We would like to thank the ARCHIE Programme Investigators and ARCHIE Programme Steering Committee for their support with producing this protocol. We would also like to thank patient representatives from the NIHR Medicines for Children Research Network Clinical Study Groups, the parents and young people who took part in our discussion groups and our patient advisers, Farrah Pradhan and Robert Newby, for their input with study design and strategies for delivery. Finally we would like to thank staff in Oxford University's Primary Care Clinical Trials Unit who were responsible for trial management, database development and assisted with trial design.

**Contributors** KW, TC and ST drafted the manuscript. KW is chief investigator of the ARCHIE trial. TC is the senior trial manager, and ST is the trial manager. MGS, ADH, MM, PL, CB, AF and AH contributed to development of the trial protocol and planning of recruitment strategies. MGS also provided input with aspects of the trial relating to microbiological assessments. MGS is the regional principal investigator (PI) of the trial in Liverpool, ADH is regional PI in Bristol and MM is regional PI in Southampton. AH is the ARCHIE programme lead. L-MY, SM and RP provided input with statistical aspects of the trial and developed the plan for statistical analysis of trial data. JW developed the economic evaluation plan and is leading the economic evaluation of trial findings. All authors have reviewed and contributed edits to the manuscript.

**Funding** The ARCHIE trial is funded by the National Institute for Health Research under its Programme Grants for Applied Research Programme (Grant Reference RP-PG-1210-12012).

**Disclaimer** The views expressed are those of the authors and not necessarily those of the UK National Health Service, the NIHR, or the UK Department of Health.

**Competing interests** KW is a National Institute for Health Research (NIHR) Postdoctoral Fellow. AF is a NIHR Senior Investigator and is supported by the Oxford NIHR Biomedical Research Centre. AH is Deputy Chairman of the Joint Committee on Vaccination and Immunisation, which advises the UK government on vaccine policy. MGS is an NIHR Investigator, is supported by the NIHR Health Protection Research Unit in Emerging and Zoonotic Infections and is a member of the New and Emerging Respiratory Viral Threats Advisory Group, whose remit includes advice to the UK government on antibiotic and antiviral stockpile policy.

**Patient consent** Not required.

**Provenance and peer review** Not commissioned; externally peer reviewed.

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
