## [Reviewer comments · BMJ Open]

ARTICLE DETAILS

TITLE (PROVISIONAL)	The early use of Antibiotics for at Risk CHildren with Influenza (ARCHIE): protocol for a double-blind randomised placebo-controlled trial
AUTHORS	Wang, Kay; Carver, Tricia; Tonner, Sharon; Semple, Malcolm G; Hay, Alastair; Moore, Michael; Little, Paul; Butler, C; Farmer, Andrew; Perera, Rafael; Yu, Ly-Mee; Mallett, Susan; Wolstenholme, Jane; Harnden, Anthony

VERSION 1 – REVIEW

REVIEWER	Maayan Gruber Galilee Medical Centre, Nahariya, Israel
REVIEW RETURNED	24-Dec-2017

GENERAL COMMENTS	Thank you for your hard work on this important subject. I am impressed with the amount of work put into this manuscript and looking forward to reading the results of this study. The protocol is well-designed and well-written and I have no specific comments or concerns.
---

REVIEWER	Tengbin Xiong IQVIA, China.
REVIEW RETURNED	30-Jan-2018

GENERAL COMMENTS	The protocol is well written and organized for a multi-center double-blind randomized placebo-controlled trial, which is to determine whether early treatment with co-amoxiclav reduces the likelihood of re-consultation due to clinical deterioration in 'at risk' children with influenza/influenza-like illness (ILI). More important, the study will evaluate potential implications of antibiotic treatment on antimicrobial resistance, and examine the cost-effectiveness, costs and effectiveness of early antibiotic use in 'at risk' children with influenza/ILI. The background, general information and literature review is comprehensive, the objectives are clear, the trial design and statistics analysis plan are sound. I only have the following questions and it would be better if they can be further clarified. 1. Although ethical approvals have been obtained and the trial has been registered, the statement that trial will be conducted in compliance with GCP is still needed.2. In study design, are there any discontinuation criteria for participants?3. In study design, could it be further clarified for maintenance of randomization codes and procedures for breaking codes?
---

	4. Are there any withdrawn subjects? If so, when and how? What are the types and timing of data to be collected from the withdrawn subjects? How to follow up withdrawn subjects? 5. How to monitor subject compliance? 6. What are the Criteria for termination of trial? 7. What procedure is for missing, unused and spurious data?
--	--

REVIEWER	Pia Hardelid UCL Great Ormond Street Institute of Child Health
REVIEW RETURNED	06-Mar-2018

GENERAL COMMENTS	This protocol describes an important trial of early antibiotics for children at increased risk of influenza complications. In particular, the 'Reflections' section will be extremely helpful for other researchers thinking of setting up similar trials in primary care. The protocol is generally very clear. I just have a few comments/queries. ILI case definition: Why did you settle on cough in addition to fever for the ILI case definition? Children may present a range of upper and lower respiratory symptoms as a result of influenza infection, in addition to fever. It would be helpful if you could clarify why this case definition was chosen, and state the sensitivity, specificity & PPV of this case definition for identifying influenza during periods when influenza is circulating. I realise you use ILI symptoms rather than influenza since children will not be tested in primary care before treatment, but there is some conflation between ILI and influenza in the protocol. Age range: Can you say why you don't include children less than 6 months old? They are much more likely to get severe influenza symptoms, particularly if they have underlying chronic conditions, and they do not qualify for influenza vaccination so they are at even high risk. Similarly, why do you stop at 12 years old? Sample size calculation: I can't see what assumption you made (if any) of the proportion of children in each gp practice would be considered at high risk? And I can't see how many practices/hospitals you estimate you needed to recruit from? Safety: I am not clear why you didn't report on the common co-amoxiclav side effects – wouldn't this have been important for your cost-effectiveness analysis, particularly if it results in another consultation? Reflection: Do you think the introduction of the universal influenza vaccination programme also hampered recruitment since clinicians were less likely to consider a vaccinated child with ILI symptoms to have influenza?
---

VERSION 1 – AUTHOR RESPONSE

Editors comments to Author:

Please revise the 'Strengths and limitations' section of your manuscript. This section should relate specifically to the methods, and should not include a general summary of, or the results of, the study. We have now revised this section to relate specifically to the methods used in our study. This section now reads:

- This randomised controlled trial specifically focuses on 'at risk' children with known risk factors for complications from influenza/influenza-like illness (ILI), who may potentially benefit most from early antibiotic treatment.

- The primary outcome (re-consultation due to clinical deterioration within 28 days) is clinically relevant and has important implications for cost-effectiveness of early antibiotic treatment.
- The study uses a pragmatic case definition of ILI, which reflects current clinical practice, but does not reliably distinguish between influenza infection and ILI due to other causes.

Along with your revised manuscript, please include a copy of the SPIRIT checklist indicating the page/line numbers of your manuscript where the relevant information can be found (<http://www.spirit-statement.org/>)

We have now included a copy of the SPIRIT checklist. Since this checklist relates to protocol documents rather than protocol manuscripts, we have indicated page numbers on both our protocol document and manuscript as appropriate which relate to the items on the checklist. We have also attached a copy of our protocol document for your reference.

Please ensure that your protocol is formatted according to our Instructions for Authors: <http://bmjopen.bmj.com/pages/authors/#studyprotocols>
We have now formatted our manuscript according to the instructions at the above link.

Reviewer 1

Thank you for your hard work on this important subject. I am impressed with the amount of work put into this manuscript and looking forward to reading the results of this study. The protocol is well-designed and well-written and I have no specific comments or concerns.

Reviewer 2

Although ethical approvals have been obtained and the trial has been registered, the statement that trial will be conducted in compliance with GCP is still needed.
We have now added the following statement to the beginning of paragraph 2 in our section on 'Ethics and dissemination': "The trial will be conducted in accordance with the principles of Good Clinical Practice (GCP) and relevant regulations."

In study design, are there any discontinuation criteria for participants?

We have now added the following text as paragraph 2 in our section on 'Safety and adverse event reporting': "Clinicians will be advised to discontinue a participant's study medication if he/she experiences an adverse drug reaction related to the study medication. Parents/guardians of participants whose study medication is discontinued will still be requested to complete their study diaries and questionnaires and will still receive telephone follow-up calls unless they choose to withdraw consent for these."

In study design, could it be further clarified for maintenance of randomization codes and procedures for breaking codes?

We have now provided further clarification regarding maintenance of randomisation codes in the paragraph 1 of our section on 'Randomisation and blinding': "The randomisation system will be implemented and managed by the Clinical Trials Unit. The randomisation codes will be maintained by a specifically appointed independent custodian."

We have also provided further clarification in relation to procedures for breaking codes in paragraph 2 of this section: "A participant's treatment allocation will be unblinded in the event of a suspected unexpected serious adverse reaction (SUSAR). Where there is a perceived need for unblinding, the participant's responsible clinician (i.e. the clinician treating the participant) will discuss the case with the Chief Investigator or a designated alternative study clinician. If it is agreed that unblinding is required, the Chief Investigator or designated study clinician will request that the randomisation code

is accessed by the independent custodian, who will disclose the participant's treatment allocation to his/her responsible clinician."

Are there any withdrawn subjects? If so, when and how? What are the types and timing of data to be collected from the withdrawn subjects? How to follow up withdrawn subjects?

The only withdrawn subjects will be those whose parents/guardians withdraw consent for follow-up data collection as described above. Follow-up data will still be collected as described in the study protocol from participants in whom study medication is discontinued unless their parents/guardians specifically withdraw consent for this.

How to monitor subject compliance?

Adherence to study medication will be monitored by asking parents/guardians to record doses of study medication given to study participants in their study diaries, as explained in paragraph 2 of our section on 'Data collection': "A parent or guardian will be asked to complete and return by post a series of four one-week study diaries on behalf of the participant to provide data on doses of study medication given to the child....".

To maximise completeness of these data, we also monitor adherence to study medication during telephone follow-ups with parents/guardians one week and two weeks after randomisation. We now also clarify this in paragraph 2 of our section on 'Data collection': "Health care professionals will also contact each participant's parent or guardian by telephone one week and two weeks after randomisation to collect data on duration of fever, adverse events, and adherence to study medication."

What are the Criteria for termination of trial?

Our manuscript already states that the Data and Safety Monitoring Committee (DSMC) will "evaluate the risk of the trial continuing and take appropriate action where necessary". This could potentially include termination of the trial. However, to make this point clearer, we have now added the following text to the end of our section on 'Safety and adverse event reporting': "The DSMC will also advise on whether the trial should be terminated based on its reviews of serious adverse events in consultation with the Trial Steering Committee and trial Sponsor if necessary."

What procedure is for missing, unused and spurious data?

We have now clarified this by amending the first sentence in our section on 'Statistical analysis and economic evaluation plan'. This now reads: "We will perform an intention-to-treat analysis including all randomised participants and use multiple imputation methods to account for missing data."

Reviewer 3

ILI case definition: Why did you settle on cough in addition to fever for the ILI case definition? Children may present a range of upper and lower respiratory symptoms as a result of influenza infection, in addition to fever. It would be helpful if you could clarify why this case definition was chosen, and state the sensitivity, specificity & PPV of this case definition for identifying influenza during periods when influenza is circulating. I realise you use ILI symptoms rather than influenza since children will not be tested in primary care before treatment, but there is some conflation between ILI and influenza in the protocol.

As far as we are aware, there is no clinical case definition of influenza-like illness (ILI) which is both accurate at identifying influenza infection and sufficiently pragmatic to be applicable across the entire target age range included in our trial (6 months to 12 years inclusive). We are aware that influenza/ILI may give rise to other upper and lower respiratory symptoms, such as dyspnoea and sputum production. However, we did not feel that including these in our ILI case definition would be

appropriate in the context of our trial, as it is not possible to reliably assess them in infants or young children. We also feel that the pragmatic ILI case definition we have used is in keeping with the pragmatic nature of our trial. We now clarify this in paragraph 1 of our section on ‘Study participants’: “...We will define ILI as the presence of both cough and fever, which may be defined as child-reported fever, parent-reported fever or temperature >37.8°C (axillary or tympanic temperature measurement). This is intended as a pragmatic case definition, which can be reliably applied across our entire target age range.”

We have also highlighted this as a limitation in our ‘Strengths and limitations’ section: “The study uses a pragmatic case definition of ILI, which reflects current clinical practice, but does not reliably distinguish between influenza infection and ILI due to other causes.”

We acknowledge that we refer to influenza/ILI in the context of children’s presentations in health care settings and include all randomised participants, irrespective of whether they have influenza or ILI, in our analysis. However, we do not feel that this constitutes conflation between influenza and ILI. Again, we feel this represents the pragmatic nature of our trial and the fact that routine clinical practice does not include testing children before treatment, as the reviewer correctly points out.

Given that we will be conducting Polymerase Chain Reaction analyses to identify trial participants who have influenza infection, we do not feel that stating sensitivity, specificity and PPV of our case definition is relevant in terms of justifying our case definition. However, for information, the table below summarises the findings of a cross-sectional study performed in primary care which developed prediction rules to distinguish influenza from ILI used an ILI case definition of cough and fever > 37.8°C (Michiels B et al. BMC Fam Pract. 2011 Feb 9;12:4. doi: 10.1186/1471-2296-12-4.):

Timing	Sensitivity	Specificity	Positive Likelihood Ratio	Negative Likelihood ratio
During epidemic	0.49 (0.40 to 0.58)	0.91 (0.89 to 0.92)	0.19 (0.16 to 0.22)	1.12 (1.08 to 1.16)
Pre/post epidemic	0.48 (0.39 to 0.57)	0.93 (0.91 to 0.95)	0.15 (0.12 to 0.17)	1.09 (1.06 to 1.12)

Age range: Can you say why you don’t include children less than 6 months old? They are much more likely to get severe influenza symptoms, particularly if they have underlying chronic conditions, and they do not qualify for influenza vaccination so they are at even high risk. Similarly, why do you stop at 12 years old?

As the reviewer mentions, children under 6 months old with chronic underlying conditions are at particularly high risk of clinical deterioration. We therefore felt that it was unlikely that clinicians would be in equipoise about whether or not antibiotic treatment would be needed in these children if they presented with an influenza-like illness. We decided to set the upper limit of our target age range at 12 years old because we were keen to focus our study on children, and felt that older children’s physiology would be very similar to that found in adults.

Sample size calculation: I can’t see what assumption you made (if any) of the proportion of children in each gp practice would be considered at high risk? And I can’t see how many practices/hospitals you estimate you needed to recruit from?

We did not make any assumptions about the proportion of children in each GP practice who would fit our study definition of ‘at risk’ because we anticipated a large degree of variation due to differences in patient list demographics and procedures for coding many ‘at risk’ conditions. It was also not possible to make any meaningful assumptions about the proportion of ‘at risk’ children in each hospital due to

variations in the types of specialist clinics provided and the opportunistic nature of recruitment from emergency department and ambulatory unit settings. However, as we explain in our section on 'Strategies for addressing challenges encountered', we were able to utilise infrastructure within the NIHR Clinical Research Networks to identify additional hospitals and general practices when recruitment proved more challenging than anticipated.

Safety: I am not clear why you didn't report on the common co-amoxiclav side effects – wouldn't this have been important for your cost-effectiveness analysis, particularly if it results in another consultation?

We will collect data on common co-amoxiclav side-effects and consider these data in our cost-effectiveness analysis. We will also publish these data in the manuscript reporting our trial results. However, we felt that it would not be informative or productive to apply formal adverse event reporting procedures (including severity and causality assessments) in the majority of these cases unless side-effects were felt to potentially compromise the safety of study participants (i.e. clinically severe or resulting in serious adverse event). This is already stated at the end of paragraph 1 in our section on 'Safety and adverse event reporting'.

Reflection: Do you think the introduction of the universal influenza vaccination programme also hampered recruitment since clinicians were less likely to consider a vaccinated child with ILI symptoms to have influenza?

Other research we have conducted as part of the wider ARCHIE programme suggests that GPs rarely consider the possibility of influenza infection specifically or ask about vaccination status when assessing children with known underlying comorbidities who present with ILI (Ashdown H et al. BMJ Open. 2016 Jun 10;6(6):e011497. doi: 10.1136/bmjopen-2016-011497). We therefore feel that this is unlikely to have been a significant factor in hampering recruitment. We have now included this point in paragraph 3 of our section on 'Reflection': "However, previous research suggests that GPs rarely consider influenza infection or ask about vaccination status when assessing 'at risk' children with ILI.(30) It is therefore unlikely that recruitment would have been significantly compromised due to health care professionals failing to consider vaccinated children as potential recruits."

VERSION 2 – REVIEW

REVIEWER	Pia Hardelid UCL Great Ormond Street Institute of Child Health
REVIEW RETURNED	19-Mar-2018
GENERAL COMMENTS	The authors have addressed my comments from the previous round.